# Sand Dust Images Enhancement Based on Red and Blue Channels

**DOI:** 10.3390/s22051918

**Published:** 2022-03-01

**Authors:** Fei Shi, Zhenhong Jia, Huicheng Lai, Sensen Song, Junnan Wang

**Affiliations:** 1School of Information Science and Engineering, Xinjiang University, Urumqi 830046, China; sigofei@xju.edu.cn (F.S.); lai@xju.edu.cn (H.L.); song_sen_sen@stu.xju.edu.cn (S.S.); 1254982138@stu.xju.edu.cn (J.W.); 2Key Laboratory of Signal Detection and Processing, Xinjiang University, Urumqi 830046, China

**Keywords:** image enhancement, RCC, sand dust images, red channel, BDPR

## Abstract

The scattering and absorption of light results in the degradation of image in sandstorm scenes, it is vulnerable to issues such as color casting, low contrast and lost details, resulting in poor visual quality. In such circumstances, traditional image restoration methods cannot fully restore images owing to the persistence of color casting problems and the poor estimation of scene transmission maps and atmospheric light. To effectively correct color casting and enhance visibility for such sand dust images, we proposed a sand dust image enhancement algorithm using the red and blue channels, which consists of two modules: the red channel-based correction function (RCC) and blue channel-based dust particle removal (BDPR), the RCC module is used to correct color casting errors, and the BDPR module removes sand dust particles. After the dust image is processed by these two modules, a clear and visible image can be produced. The experimental results were analyzed qualitatively and quantitatively, and the results show that this method can significantly improve the image quality under sandstorm weather and outperform the state-of-the-art restoration algorithms.

## 1. Introduction

Images or videos captured in sandstorm scenes usually have low contrast, poor visibility and yellowish tones. This is because sand dust particles scatter and absorb specific spectra of light between the imaging devices and the observed objects. Therefore, these degraded sand dust images will greatly lose their quality and degrade the performance of computer vision application systems that typically work outdoors during inclement weather conditions. Such systems include video surveillance systems for public security monitoring [1,2], intelligent transportation systems for license plate recognition [3,4], visual recognition systems for automatic driving [5], and so on. Hence, developing an effective sand dust image restoration method to restore color and contrast for computer vision application systems is desirable. To improve the performance of computer vision systems and restore the visibility of degraded images, some restoration algorithms for degraded sand dust images have been proposed. Huang [6] presented a transformation method that enhances the contrast of degraded images via the gamma correction technique and probability distribution of bright pixels. AlRuwaili [7] proposed an enhancement scheme, the degraded input image is first converted into an HIS color space, and then color cast corrections and contrast stretching are performed. Zhi [8] restored vivid sand dust images by using color correction, SVD and the contrast-limited adaptive histogram equalization algorithm. Tri-threshold fuzzy operators are introduced to enhance contrast by Al-Ameen [9]. Yan [10] enhanced dust images by improving the sub-block partial overlapping histogram equalization algorithm. Shi [11] enhanced images by combining contrast limited adaptive histogram equalization (CLAHE) and gray world theory. Tensor least square method is proposed to enhance sand dust image by Xu [12]. Cheng [13] using white balance and guided filtering technologies. Park [14] proposed a Coincidence histogram. Although the above traditional algorithms have some effects on the restoration of sand dust images, the restored images appear over-enhanced or under-enhanced, and the color is distorted.

In order to provide better visual quality of degraded sand dust images, several visibility restoration methods using the atmospheric transmission model have been presented. Yu [15] introduced a method for restoring single sand dust image that depends on using atmospheric transmission model and constraining information loss. The atmospheric light is first estimated using the grey-world assumption and the scattering model, then, a fast guide filter is used to suppress the halos in the post process. Wang [16] first considered multiple scattering factors, then, particle swarm optimization method was used for optimizing the exposure parameters and the atmospheric light in order to obtain better restored images. Peng [17] assumed that the ambient light was known; then, images were restored by calculating the difference between the light intensity observed in the degraded image scene and the ambient light intensity. Huang [18] presented a novel Laplacian-based image restoration method. The minimum filter and Gauss adaptive transform are introduced by Yang [19]. Kim [20] proposed a method based on saturation transmission estimation. However, the above-mentioned methods caused the processed image to appear blocky, haloed or over-enhanced, and the methods cannot handle sand dust images that contain heavy yellow tones.

Image dehazing algorithms have attracted great attention in recent years. One attractive solution for image dehazing is the neural network approach, for instance, Cai [21] designed a novel Ranking-CNN for single image dehazing. Yang [22] proposed a Region Detection Network model, which reflects the regional detection of a single hazy image. Li [23] proposed a PDR-Net for single image dehazing. Their findings show that a neural network can better estimate ambient light and the transmission than other approaches. However, training a neural network needs a lot of datasets, and there is no dataset for sand dust images, so the neural network is not suitable for sand dust images processing at present. Another effective solution for image dehazing is the dark channel prior (DCP) method raised by He [24]; this is a natural image-based observation that one of the RGB channels has very low intensity for most pixels. The DCP method is very useful for haze removal, and its calculation is simple. Many improved DCP methods have been applied in various fields, such as image dehazing [25,26], underwater image enhancement [27,28], and sand dust image restoration [29,30,31,32,33].

To effectively remove atmosphere particles from the sand dust images, Fu [29] proposed a restoration method for sand dust images by using fusion strategy. The input sand dust image was color-corrected by using the statistical scheme first. Then, gamma correction technology with two different coefficients and DCP were applied. Finally, the input images and the weight maps are fused to obtain enhanced image. Peng [30] proposed a general dark channel prior method to restore sand dust, haze and underwater images. He estimated the ambient light by adopting a depth-related color change. Then, he calculated the difference between ambient light and scene intensity. Shi [31] proposed a DCP method for enhancing sand dust images and reducing halos. The method included three modules: The color casting was first corrected by using gray world theory in LAB space; then, sand particles were removed by an improved DCP-based dehazing method; finally, a gamma function was used to stretch contrast. Gao [32] proposed the method of reversing blue channel. Cheng [33] combined white balance and reversing blue channel technology to enhance sand dust images. However, the improved DCP algorithms mentioned above create block artifacts, color distortion and yellow tones when restoring degraded images taken under sandstorm weather.

In this paper, we proposed a method for restoring sand dust images by using red-blue channels, which takes the advantages of the proposed red channel correction function (RCC) module and the blue channel dust particles removal (BDPR) module. By combining them, the color deviation problems and the underestimation of scene depth can be effectively overcome. Compared with the other improved DCP algorithms, our algorithm is founded upon the imaging characteristics of the sand dust images. By adopting this scheme, the proposed algorithm can effectively generate clear images. In a word, the contributions of the paper are reflected in the following three aspects:The red channel correction function (RCC) can effectively avoid the problem of insufficient or excessive color cast adjustments in real sand dust images. It restores the lost color channel from the other channels. Because the dust particles absorb less of the red ray under the dusty weather conditions, causing the red ray decay to be the slowest.After the input image is processed by the correction function, the blue channel dust particles removal (BDPR) module is applied to remove atmospheric particles in degraded images. We assume that the dust particles absorb blue rays quickly; hence, the intensity of the blue channel is lower. The proposed method can remove sand dust particles more effectively, eliminating the blueish tone of the restored image.To obtain more accurate transmission and atmospheric light, the sand dust image and the corrected image are applied to the BDPR module simultaneously, where the sand dust image is used in atmospheric light estimation, and the corrected image is used for calculating transmission.

The rest of the paper is organized as follows. Section 2 reviews the dark channel prior method. Section 3 introduces the proposed in this paper methods and algorithms in detail. Section 4 introduces experimental results of this method and state-of-the-art sand dust image restoration algorithms and analyzes them in detail. Finally, Section 5 summarizes the advantages and limitations of the proposed method and suggests directions for further research.

## 2. Background

In this section, we will briefly review the dehazing method based on the dark channel prior [24], which has been widely improved and applied in the restoration of hazy, underwater and sandstorm images.

In the fields of computer graphics and computer vision, imaging models are widely used, which describes the light scattering and absorption between the camera and the observation scene, shown in Figure 1.

Assuming that the light decay is homogeneous, the formation model for a hazy image is given by [34]:(1)$$D^{c}\left(x\right)=R^{c}\left(x\right)*t\left(x\right)+A^{c}*\left(1-t(x)\right),c\in \left\{r,g,b\right\}$$
where $D^{c}\left(x\right)$ is the intensity of the *c* channel where the color image is observed at *x* pixel, $R^{c}\left(x\right)$ is the intensity of the haze-free scene, t(x) represents the medium transmission, $A^{c}$ is the ambient light, *c* indicates one of the RGB color channels, and the value ranges of $D^{c}\left(x\right)$, $R^{c}\left(x\right)$ and $A^{c}$ are set to [0, 1].

The DCP is based on the observations of outdoor haze-free images, which show that approximately 75% of the pixels of the non-sky area have an intensity such that one of the RGB color channels is zero. The DCP is as follows:(2)$$R_{\mathrm{Dark}}^{rgb}\left(x\right)=\min\left(\min_{y\in \Omega (x)}\;\left(R^{R}\left(y\right)\right),\min_{y\in \Omega (x)}\;\left(R^{G}\left(y\right)\right),\min_{y\in \Omega (x)}\;\left(R^{B}\left(y\right)\right)\right)\approx 0$$

Implementing the minimum operators on the local patch of the RGB channels in Equation (Equation 1) and dividing both sides of Equation (Equation 1) by $A^{c}$, the transmission t(x) can be roughly calculated as:(3)$$t\left(x\right)=1-\omega \min_{y\in \Omega (x)}\;\;\;\;\left\{\;\;\;\min_{c\in \{r,g,b\}}\;\frac{D^{c}\left(y\right)}{A^{c}}\;\;\;\right\}\;\;\;$$
where ω=0.95 to leave some haze in the restored scene brightness to make it looks natural. Because a block artifact is generated by using local minimum filtering, it can be improved by using the soft matting method [35] or by a guided filter method [36]. Atmospheric light $A^{c}$ is chosen from the brightest 0.1% of the pixels based on the DCP for the hazy image.

Finally, according to the formation model for hazy images expressed in Equation (Equation 1), by solving the image formation process inversely, the restored image $R^{c}\left(x\right)$ is calculated as:(4)$$R^{c}\left(x\right)=\frac{D^{c}\left(x\right)-A^{c}}{\max(t\left(x\right),t_{0})}+A^{c},c\in \;\;\;\left\{\;\;\;r,g,b\;\;\;\right\}\;\;\;$$
where $t_{0}$ is an empirical parameter, which is set to 0.1 to improve the exposure of scene radiance.

## 3. Proposed Algorithm

The image formation model in Section 2 suggests that the estimations of the medium transmission and atmospheric light are very important to restore degraded images. However, the inherent features of the degraded sand dust image make the traditional dehazing algorithms, based upon the atmospheric propagation model for hazy images, unable to process the images with color cast. To this end, we propose a sand dust removal method and algorithms based on red and blue channels to recover the visual and color effects. The proposed method consists of both an RCC module and a BDPR module for which novel algorithms are proposed here. The flowchart of our method is shown in Figure 2.

First, the original sand dust image is corrected by the red channel correction function module to overcome the problem of yellow or red color deviation in sand dust images. Second, the blue channel sand dust removal module is applied to restore image details and remove atmospheric particles, which is based on the characteristic that the blue ray is quickly absorbed during sandstorm weather conditions. However, the image processed by module BDPR is darker and has some blue tones. Thus, we use the contrast enhancement method in [37,38] to increase the contrast of the color-corrected image. Finally, the clear restored image is generated by fusing the enhanced image with the sand-dust-removed image using wavelet fusion technology.

### 3.1. RCC Module

For image color correction, gamma correction technology [6] and grayscale world theory [7] are widely used. However, sand dust images have serious color casting because the green and blue rays in the atmosphere are absorbed and scattered by sand dust particles. The gamma correction and gray world hypothesis approaches cannot be applied to correct color casting directly, which may result in color distortion and image over-enhancement.

The color casting of sand dust images is caused by the light attenuation. The red ray decay is slowest, and the blue light is absorbed fastest under sandstorm weather, and the RGB channel histogram of the degraded image is sequential. Two histograms of the original sand dust images are shown in Figure 3, where images on the left are sand dust images, and right side images are the histograms of the RGB channels.

Based on the above observation, we propose a color correction function module, which depends on the red channel to effectively adjust the image. First, the histogram of the red channel is used as a reference in the RCC module, and the histograms of the blue and green channels are translated as follows:(5)$$\left\{\begin{array}{c} Ir(x)=Ir(x)\\ Ig(x)=Ig(x)+(\mu_{r}-\mu_{g})\\ Ib(x)=Ib(x)+(\mu_{r}-\mu_{b}) \end{array}\right.$$

Then, color is stretched in RGB color space, which is described as follows:(6)$$\left\{\begin{array}{c} I_{c}^{\max}=\mu_{c}+k*\sigma_{c}\\ I_{c}^{\min}=\mu_{c}-k*\sigma_{c}\\ I_{c}=255*\left(I_{c}-I_{c}^{\min}\right)/\left(I_{c}^{\max}-I_{c}^{\min}\right) \end{array}\right.$$
where $\mu_{c},c\in \left\{r,g,b\right\}$ is the mean value of each channel in RGB color space, $\sigma_{c},c\in \left\{r,g,b\right\}$ represents the mean variance of RGB channel, and k is the adjustment factor. k is set to 2 in our experiment. Images are processed by RCC module as shown in Figure 4. It is easy to see that the method is simple and effective for correcting sand dust images, but the corrected image is still blurry.

### 3.2. BDPR Module

The first term, $R^{c}\left(x\right)*t\left(x\right),c\in \;\;\;\left\{\;\;\;r,g,b\;\;\;\right\}\;\;\;$, in Equation (Equation 1) is the direct decay component, and $A^{c}*\left(1-t\left(x\right)\right),c\in \left(r,g,b\right)$ is atmospheric light attenuation. The direct decay component describes the radiation and attenuation of the scene in the medium, while atmospheric light attenuation describes the scene changes caused by light scattering. Moreover, the Equation (Equation 1) shows that the radiant light of the scene first goes through the multiplier attenuation, and then through the additive attenuation.

According to the Beer Lambert law, the propagation of light decreases exponentially with increasing distance. Assuming the atmosphere is uniform, the medium transmission map t(x) can be indicated as:(7)$$t\left(x\right)=e^{-\beta d(x)}$$
where d(x) is the scene depth, and β is the atmospheric scattering coefficient. As d(x) approaches 0, t(x) approaches 1, so the atmospheric light attenuation cannot be affected. In contrast, when d(x) is not 0, t(x) decreases as d(x) increases, and atmospheric light attenuation takes a dominant role. However, Equation (Equation 1) cannot be used directly for sand dust images; hence, we transform Equation (Equation 1) into Equation (Equation 8) to take advantage of the inherent characteristic that the intensity of blue channel in color sand dust image is very low:(8)$$\left\{\begin{array}{c} D^{c}\left(x\right)=R^{c}\left(x\right)*t\left(x\right)+A^{c}*\left(1-t\left(x\right)\right),c\in \left\{r,g\right\}\\ 1-D^{b}\left(x\right)=1-R^{b}\left(x\right)*t\left(x\right)+1-A^{b}*(1-t\left(x)\right) \end{array}\right.$$
where $D^{c}$ and $R^{c}$ represent the degraded sand dust image and the original image, respectively. Please note that Equation (Equation 8) is equivalent to Equation (Equation 1). Therefore, Equation (Equation 8) reflects the fact that the light decays with distance, which actually happens in sandstorm weather. The only difference we have to account for is that blue intensity attenuates faster as distance increases. Hence, we modified the DCP method according to [27,32]. It states that:(9)$$\begin{array}{c} R^{Blue}\left(x\right)=\min\left(\min_{y\in \Omega (x)}\;\left(R^{R}\left(y\right)\right),\min_{y\in \Omega (x)}\;\left(R^{G}\left(y\right)\right),\min_{y\in \Omega (x)}\;\left(1-R^{B}\left(y\right)\right)\right)\approx 0 \end{array}$$
for a non-degraded sand dust image, where ω(x) represents the neighborhood pixels around the pixel *x*. Note that in the degraded image near the observer, the blue channel intensity is high, so its reciprocal $1-R^{B}\left(y\right)$ is low, and the prior is true. However, blue intensity rapidly attenuates as the distance increases, so the prior starts to become false. This fact will help to estimate the depth map of the scene and the atmospheric light.

Restoration for a single degraded sand dust image using Equation (Equation 9) is a very challenging task, because there is little image information available, and the estimation accuracy of the medium transmission and atmospheric light is related to the recovery quality of the image.

In previous studies, atmospheric light was chosen to be the most opaque area of haze in the image. In this paper, the atmospheric light is estimated by Equations (Equation 8) and (Equation 9) using the input degraded sand dust image. In addition, the brightest pixel of 0.1% is selected as the atmospheric light estimation, as suggested in [24]. Figure 5 shows the atmospheric light position selected by the proposed algorithm, and the dark channel prior algorithms in two images.

The red area in Figure 5 is the estimated position of atmospheric light. It is thus clear that the brightest area in the sky is selected by the dark channel algorithm, while the proposed algorithm chose the most opaque sand dust area, not the brightest sky area or other white objects (such as the white cars in the picture) as the atmospheric light. This shows that the proposed method is better at choosing atmospheric light.

After the atmospheric light is estimated, another key step is to calculate medium transmission. Since the degraded sand dust image has low contrast and color distortion, it cannot be applied to estimate transmission. We use the bright and dark channels of the corrected image, which are processed by the RCC module, and their difference to estimate the transmission [17]. This method assumes that the density of the sand dust image is related to the maximum and minimum of the channels and their difference, which is defined as: (10)$$\begin{array}{c} \left\{\begin{array}{c} d\propto I_{bright}\propto \left(I_{bright}-I_{dark}\right)\\ I_{dark}\left(x\right)=\min\left\{I_{r}^{RCC}\left(x\right),I_{g}^{RCC}\left(x\right),1-I_{b}^{RCC}\left(x\right)\right\}\\ I_{bright}\left(x\right)=\max\left\{I_{r}^{RCC}\left(x\right),I_{g}^{RCC}\left(x\right),1-I_{b}^{RCC}\left(x\right)\right\} \end{array}\right. \end{array}$$
where $I_{r}^{RCC}\left(x\right)$,$I_{g}^{RCC}\left(x\right)$ and $I_{b}^{RCC}\left(x\right)$ represent the RGB channels of the corrected image by the RCC module. Then, sand dust density is expressed as:(11)$$d\left(x\right)=\min_{\Omega (x)}\;\left(I_{\mathrm{dark}}(x)*\left(1-\frac{I_{bright}(x)-I_{dark}\left(x\right)}{\max(1,I_{bright}\left(x\right))}\right)\right)$$

Assuming medium transmission is locally homogeneous and is inversely proportional to d(x), medium transmission t(x) is estimated as:(12)t(x)=1−ωG(d(x))
where G(y) is the function of image guided filtering [36], and ω is a parameter for retaining the naturalness of the restored image, which is set to be 0.95 in the paper.

Finally, after the atmospheric light $A^{c}$ and transmission t(x) are calculated, the sand dust-free particle image is obtained by Equation (Equation 4).

## 4. Experimental Results

In this section, we will make qualitative and quantitative assessments on the proposed method in the paper. We compare the proposed method with 10 other state-of-the-art image restoration algorithms, including Tri-threshold fuzzy Operators (TFO) [9], normalized gamma transformation (NGT) [11], blue channel compensation and guided Image filtering (BCGF) [13], airlight white correction (AWC) [17], visibility restoration of single image (VRSI) [19], saturation-based transmission map estimation (SBT) [20], fusion-based enhancing approach (FBE) [29], generalization of the dark channel Prior (GDCP) [30], halo-reduced dark channel Prior (HDCP) [31] and as reversing the blue channel prior (RBCP) [32], whose source codes are provided by the authors. The experimental results include three parts. The first part will qualitatively discuss the restoration results of the captured images in sandstorm weather conditions. In the second part, the three evaluation methods in [39], the natural image quality evaluator (NIQE) [40], the distortion identification-based image verity and integrity evaluation (DIIVINE) index [41] and natural scene statistics and Perceptual characteristics-based quality index (NPQI) [42] are used to quantitatively analyze the restoration results of the presented algorithm and the 10 state-of-the-art algorithms. The third part will analyze the execution time of algorithms. All algorithms use MATLAB code except SBTME and are run on a computer with 2.7GHZ CPU, Intel Core i5 and with 32G RAM.

### 4.1. Qualitative Assessment

Figure 6 and Figure 7 show the restoration results of the presented method and the known benchmark methods on weak sand dust images and various sandstorm images. As shown in Figure 6, TFO [9], BCGF [13], AWC [17] and GDCP [30] do not eliminate color shift. NGT [11] and VRSI [19] cannot effectively remove sand dust. FBE [29] failed to process sand-dust images. FBE [29] can eliminate the undesirable color cast effects, but the restored image is dark, and the details are lost. HDCP [31] enhances the contrast of the image, but the contrast is over-enhanced and the restoration results are severely distorted. The image obtained by RBCP [32] is dark and blue.

For the sandstorm images shown in Figure 7, TFO [9], SBT [20], GDCP [30] and RBCP [32] obtain poor image enhancement effect. NGT [11] does not remove sand-dust particles in the image. BCGF [13], AWC [17] and HDCP [31] can not remove the color veils of sandstorm images. VRSI [19] overenhanced sand-dust images.The restored image by FBE [29] is dark and sand dust do not effectively removal.

Compared with the above 10 state-of-the-art methods, our method removes the color cast using the RCC module, and it removes sand dust particles using the BDPR module. our restored results are more natural in color, clearer in detail and more similar to real images.

### 4.2. Quantitative Assessment

In general, the objective evaluation mechanism is used to quantify the accuracy of restoration results. Because there is no clear sand dust-free reference image, it is very difficult to analyze the restored sand dust image quantitatively. Therefore, to better quantitatively evaluate the performance of this method in the processing of sand dust images, the paper uses a non-reference method and introduces the following three well-known metrics proposed in [39]: visible edges recovery percentage *e*, the saturation σ and the contrast restoration percentage $\bar{r}$. In addition, the natural image quality evaluator (NIQE) as suggested in [40], the distortion identification-based image verity and integrity evaluation (DIIVINE) index in [41] and natural scene statistics and Perceptual characteristics-based quality index (NPQI) in [42] are adopted. In the above metrics, if *e* and σ are approximately zero, that suggests a better performance, and a greater $\bar{r}$ implies that the contrast of restored image is stronger. The smaller the NIQE is, the better the quality of the restored image is. The lower the DIIVINE is, the better the distorted image quality is. The smaller the NPQI is, the better the quality of the restored image is.

The above well-known metrics are used to quantitatively evaluate the restoration performance of 12 sand dust images from Figure 6 and Figure 7 using the proposed algorithm and the ten state-of-the-art algorithms compared in this paper. The result was shown in Table 1. Compared with the other 10 state-of-the-art algorithms, the sand dust images are restored by the proposed method achieved a greater *e* and σ is closer to 0, and the obtained $\bar{r}$ is among the top ranked. Meanwhile, the proposed method can obtain better scores on DIIVINE and NPQI metrics. The experimental results in Table 1 show that the proposed in the paper method can achieve better performance in the restoration of the sand dust images, and the restored sand dust images have better performance in contrast, tones and saturation. Figure 6l and Figure 7l also demonstrates that the image restored by the proposed algorithm has obtained good effects.

In order to further verify the performance and robustness of the proposed method, we used 375 sand dust images collected from the Internet. The average scores of the six metrics on the images restored using the proposed method and compared methods are listed in Table 2. It can be seen from the Table 2 that the *e* obtained by BCGF [13] and FBE [29] is greater than that of the proposed algorithm in this paper, but other results are lower than that of the algorithm in this paper. Although the resulting σ and $\bar{r}$ of HDCP [31] is better than that obtained by the method in this paper, the actual restoration effect of HDCP [31] is obviously worse than that of the proposed algorithm. Moreover, compared with other methods, the proposed algorithm can obtain better NIQE results. As can be concluded from Table 2, it is not surprising that the method in this paper achieves the top rank DIIVINE scores and the best NPQI scores for 375 sand dust images, which is mainly due to the corrective ability inherited from RCC and BDPR modules. Through processing the results of a large number of sand dust image data sets, it indicates that the proposed method in this paper can obtain better performance in the restoration of sand dust images.

### 4.3. Running Time

Table 3 lists average run-time of different methods implementing 20 execution rounds on the real-world sand dust image of different sizes. The experiment is implemented under the Windows 10 environment of Intel i5 CPU and 32G RAM. Except for the code of SBT [20], the other codes are written in MATLAB, and all codes are provided by the author. To ensure the fairness of the comparison, SBT [20] is not included in the comparison due to its code being written in Python. As shown in Table 3, the proposed method has a higher time cost, so the application of this method to real-time systems needs to be improved. The use of guided filtering and local filtering in this method leads to the high time cost. Inspired by Kim [20], using a pixel-by-pixel compensation approach to estimate the transmission can significantly reduce the time cost and satisfy the application of real-time video image processing.

## 5. Conclusions

In this paper, a single sand dust image restoration algorithm is proposed, and it can effectively recover the sand dust images. First, according to the characteristics of the slowest attenuation of red light in the degraded image, color correction is performed using a red channel-based correction algorithm. Then, an improved DCP algorithm is improved based on the blue channel to remove sand dust particles by using the characteristic that blue light decays fastest. Through the quantitative and qualitative analysis of the restoration results of a large number of degraded sand dust images with different scenes and color casting, the proposed algorithm displays satisfactory performance in processing most of the sand dust images and can obtain reasonable restoration results. However, the proposed algorithm has the disadvantage of high time costs. In the future, we will further study more effective sand dust restoration algorithms to meet the needs of real-time vision application systems. Another further goal will be to develop methods for sand dust video restoration using spatio-temporal data modeling [43].

## Figures and Tables

**Figure 1 sensors-22-01918-f001:**
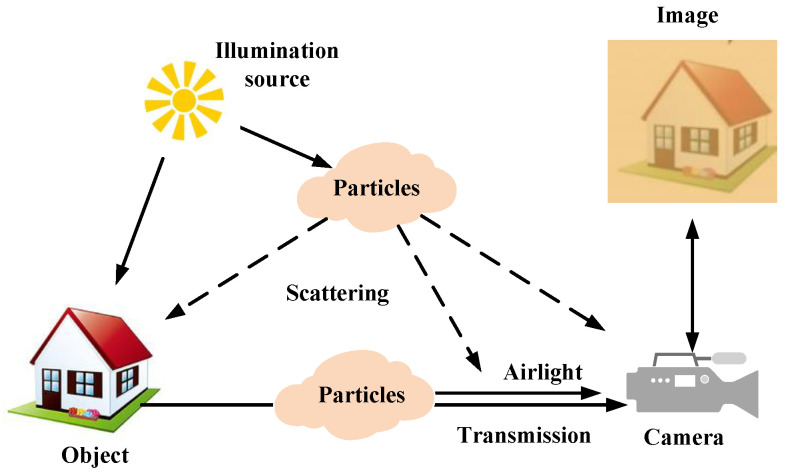
Formation model for degraded images.

**Figure 2 sensors-22-01918-f002:**
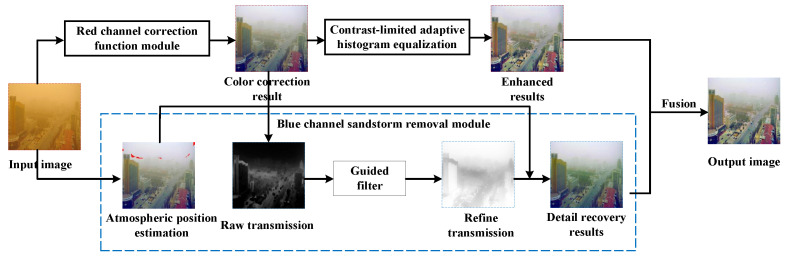
Flowchart of sand-dust image restoration method based on red and blue channel.

**Figure 3 sensors-22-01918-f003:**
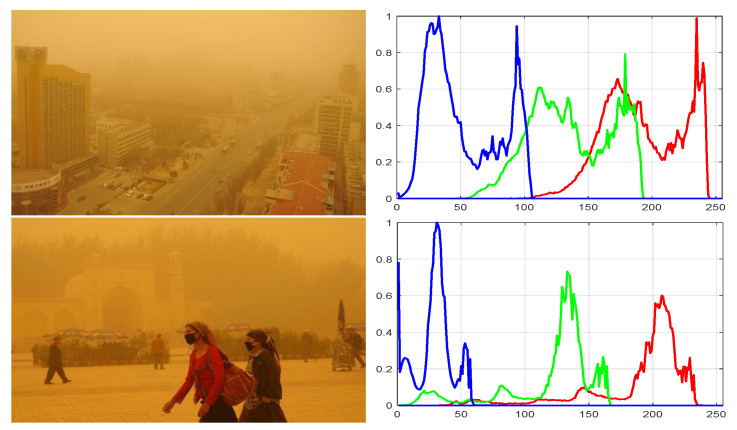
Sand dust images and histograms.

**Figure 4 sensors-22-01918-f004:**
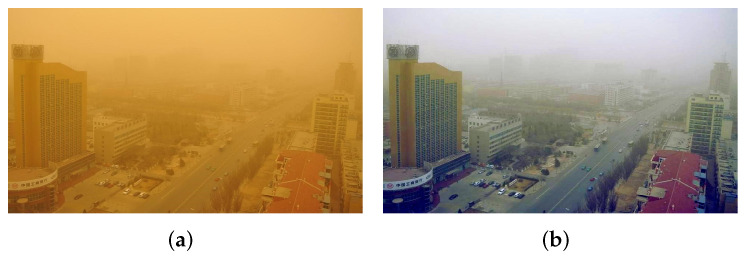
Sand dust image correction based on the red channel correction function: (**a**) Sand-dust images; (**b**) Corrected image.

**Figure 5 sensors-22-01918-f005:**
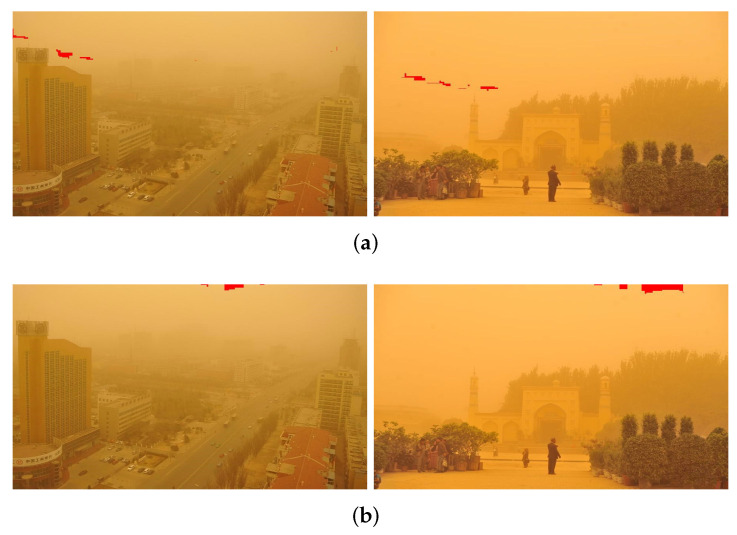
Atmospheric light position selected by two algorithms: (**a**) Proposed algorithm; (**b**) Dark channel prior algorithm.

**Figure 6 sensors-22-01918-f006:**
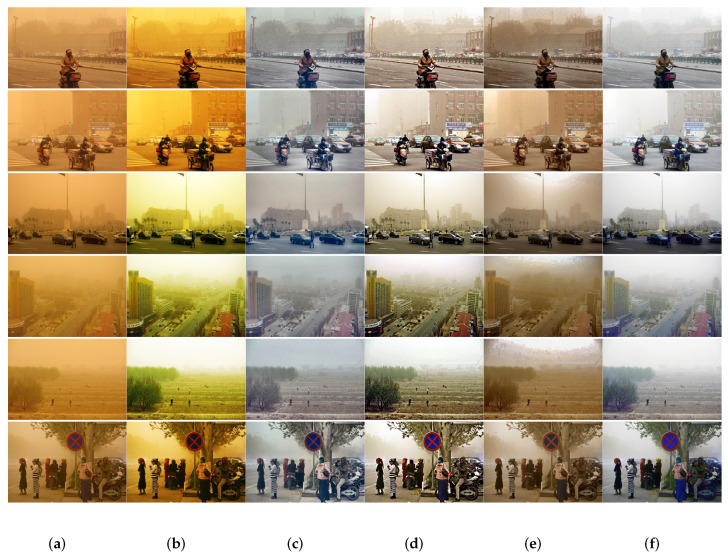
Qualitative comparison results of sand dust images with weak color cast. (**a**) Sanddust Images; (**b**) TFO [9]; (**c**) NGT [11]; (**d**) BCGF [13]; (**e**) AWC [17]; (**f**) VRSI [19]; (**g**) SBT [20]; (**h**) FBE [29]; (**i**) GDCP [30]; (**j**) HDCP [31]; (**k**) RBCP [32]; (**l**) Proposed.

**Figure 7 sensors-22-01918-f007:**
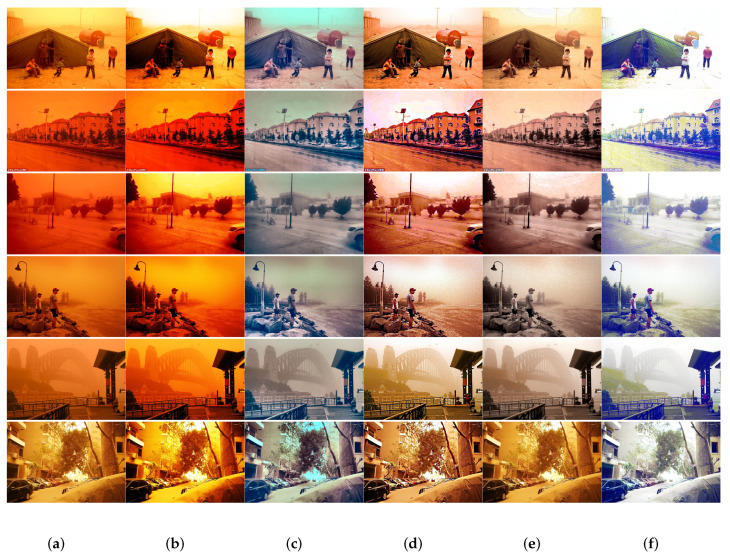
Qualitative comparison results of various sand storm images: (**a**) Sanddust Images; (**b**) TFO [9]; (**c**) NGT [11]; (**d**) BCGF [13]; (**e**) AWC [17]; (**f**) VRSI [19]; (**g**) SBT [20]; (**h**) FBE [29]; (**i**) GDCP [30]; (**j**) HDCP [31]; (**k**) RBCP [32]; (**l**) Proposed.

**Table 1 sensors-22-01918-t001:** Average results of non-reference evaluation of 12 sand dust images.

Method	*e*	σ	$\bar{r}$	NIQE	DIIVINE	NPQI
TFO [9]	0.4268	0.0693	1.5123	3.5329	32.6236	10.3987
NGT [11]	0.4268	0.0693	1.5123	3.3223	26.7959	9.5502
BCGF [13]	0.8281	0.3943	2.8015	3.3952	29.9300	9.3376
AWC [17]	0.8281	0.3943	2.8015	3.3859	27.5216	9.8819
VRSI [19]	0.4074	0.0002	2.0769	3.445	31.4305	9.8026
SBT [20]	0.7134	0.0011	1.6713	3.4191	27.1865	11.7016
FBE [29]	0.9944	0.3065	2.1572	3.3427	28.7320	9.6831
GDCP [30]	0.7125	0.0132	1.5251	3.4118	30.1233	10.9681
HDCP [31]	0.7485	0.0054	4.4502	3.6401	27.6498	10.2809
RBCP [32]	0.9136	0.0023	1.4073	3.6153	31.9274	12.3408
**Proposed**	**0.7808**	**0.0231**	**2.1968**	**3.311**	**27.6903**	**9.5006**

**Table 2 sensors-22-01918-t002:** Average results of non-reference evaluation of 375 sand dust images.

Method	*e*	σ	$\bar{r}$	NIQE	DIIVINE	NPQI
TFO [9]	1.7826	0.0611	1.7884	3.8503	35.0397	11.3340
NGT [11]	0.8204	0.00001	1.9330	3.7331	26.5634	11.1022
BCGF [13]	2.9887	0.6527	3.1582	3.7324	26.5031	10.8567
AWC [17]	1.9980	0.1666	1.5084	3.9112	27.5216	12.6198
VRSI [19]	1.3441	0.1070	1.7008	3.8898	33.4292	11.7494
SBT [20]	2.1681	0.0038	1.8638	3.7687	29.7283	11.9398
FBE [29]	2.6453	0.221	2.3218	3.7060	26.5445	10.7949
GDCP [30]	1.7376	0.1066	1.7405	3.8393	29.3818	12.1313
HDCP [31]	2.2070	0.0566	4.6496	4.0680	24.8841	11.6375
RBCP [32]	1.3951	0.1299	1.6007	3.9928	34.2842	12.3572
**Proposed**	**2.4519**	**0.0780**	**2.3671**	**3.7154**	**25.3551**	**10.7368**

**Table 3 sensors-22-01918-t003:** The running times of various methods (unit: second).

Method	500 × 300	640 × 480	1200 × 800	2000 × 1500	3648 × 1824
TFO [9]	0.0416	0.0977	0.4166	2.6107	9.1139
NGT [11]	0.6134	0.7759	1.3610	3.4490	7.1490
BCGF [13]	0.3669	0.5213	1.3794	4.1963	9.5850
AWC [17]	0.3729	0.5819	2.5781	34.183	61.833
VRSI [19]	0.6609	1.4398	4.6200	13.978	31.675
FBE [29]	1.1384	1.6348	3.6335	10.166	21.727
GDCP [30]	2.4017	4.3347	13.602	36.702	89.085
HDCP [31]	4.5432	7.8935	24.229	72.765	165.34
RBCP [32]	0.7722	1.5918	6.8674	35.4307	151.25
**Proposed**	**1.0625**	**1.6491**	**4.8029**	**13.307**	**32.021**

## Data Availability

The data presented in this study are available on request from the corresponding author. Data are not publicly available due to privacy considerations.

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
