# Peer review of "Sand Dust Images Enhancement Based on Red and Blue Channels"

_sensors, 2022, doi:10.3390/s22051918_

Round 1

Reviewer 1 Report

This paper proposed a sand dust image restoration algorithm. The proposed method is interesting and promising. I have the following concerns.

1 There is a grammatical error in the FIRST sentence of the article.
2 Authors list 3 contributions in the Introduction section. I have one question regarding the second contribution, wherein the authors state "we assume that the dust particles absorb blue rays quickly; hence...". Such an assumption is not appropriate or convincing. Do you have any reference/theory to support this assumption? If your assumption is not widely accepted, the design of the proposed method will be questionable then. Please clarify the second contribution.
3 It's better to say Section 2 and section 3 rather than the second section and the third section.
4 The histogram in Fig3 is not clear. Please increase the size of axes labels.
5 Since the proposed algorithm involves image quality assessment(IQA), apart from NIQE, I suggest adding the following IQA metrics to the Introduction section.

[1] Blind Image Quality Assessment: From Natural Scene Statistics to Perceptual Quality, TIP 2011

[2] dipIQ: Blind Image Quality Assessment by Learning-to-Rank Discriminable Image Pairs, TIP 2017

[3] Blind Image Quality Assessment by Natural Scene Statistics and Perceptual Characteristics, ACM TOMM 2020

Author Response

Dear reviewer,  I have attached the response letter, please download and review it.

Reviewer 2 Report

The paper describes a specific application of Images Enhancement.

Analysis of the results does not show a clear improvement in the method proposed compared to the others.

Given the peculiarity of the application, in the comparison must also be presented the processing time and say how you plan to reduce them.

It would also be interesting to evaluate the effects of the algorithm on images without sand dust.

Author Response

Dear reviewers, I have attached the response letter, please download and review it.

Round 2

Reviewer 1 Report

The authors have addressed all the concerns